# Daily Headache in Chronic Migraine Is a Predictive Factor of Response in Patients Who Had Completed Three Sessions of OnabotulinumtoxinA

**DOI:** 10.3390/toxins13060432

**Published:** 2021-06-21

**Authors:** Enrique Martínez-Pías, Ángel L. Guerrero, Álvaro Sierra, Javier Trigo, David García-Azorín

**Affiliations:** 1Headache Unit, Neurology Department, Hospital Clínico Universitario de Valladolid, 47003 Valladolid, Spain; enriquemartinezpias@gmail.com (E.M.-P.); alvarosierramencia@gmail.com (Á.S.); javiertrigolopez@gmail.com (J.T.); davilink@hotmail.com (D.G.-A.); 2Institute for Biomedical Research of Salamanca (IBSAL), 37007 Salamanca, Spain; 3Department of Medicine, University of Valladolid, 47005 Valladolid, Spain

**Keywords:** botulinum toxin type A, headache disorders, chronic daily headache, migraine

## Abstract

OnabotulinumtoxinA is one of the main preventive treatments for chronic migraine. Despite that up to one third of patients with chronic migraine suffer from daily headache, these individuals have hardly been studied. We conducted a prospective cohort study, including patients with chronic migraine and treated with OnabotulinumtoxinA according to the PREEMPT paradigm. The primary endpoint was to assess whether patients with chronic migraine and daily headache had a different response after three sessions of OnabotulinutoxinA than patients without daily headache. The secondary endpoint was to analyse the presence of predictive factors that could be associated with a higher response to OnabotulinumtoxinA. Patients with daily headache had a reduction of 14.9 (SD: 9.7) headache days per month, patients with 22–29 headache days a reduction of 10.6 (SD: 9.9) days, and patients with 15–21 headache days a reduction of 8.6 (SD: 7.1) days (*p* < 0.001). In the univariate regression analysis, a higher number of headache days per month at baseline was associated with higher odds of reduction in the number of headache days per month after OnabotulinumtoxinaA treatment (OR: 0.474, 95% CI: 0.278–0.670, *p* < 0.001). This association was maintained in the multivariate regression analysis (OR: 0.540, 95% CI: 0.333–0.746, *p* < 0.001). In our sample, daily headache was not associated with a worse response to OnabotulinumtoxinA treatment. A higher frequency of headache at baseline was a predictor of better response to OnabotulinumtoxinA treatment.

## 1. Introduction

Migraine is the second cause of years lived with disability between 15 and 49 years [1], which is higher in patients with chronic migraine (CM) [2]. Migraine treatment can improve a patient’s life by reducing headache frequency, intensity, and duration of attacks, and the frequency of acute symptomatic treatments use [3].

OnabotulinumtoxinA (OnabotA) [4] is one of the main preventive treatments for CM. It is effective, safe, well-tolerated [5,6], and has a cumulative effect [7]. In Spain, OnabotA is indicated in patients with inadequate response or tolerance to two oral preventive drugs [8].

Between 17–33% of CM patients suffer from daily headache [9,10]. Patients with CM and daily headache have more headache hours per day and higher disability [11]. However, patients with CM and daily headache have been excluded from clinical trials, both for OnabotA [5,12,13] and monoclonal antibodies against calcitonin gene-related peptide [14,15,16,17], and therefore, information about their response to treatment is scarce.

In the present study, we evaluated whether patients with CM and daily headache have a different response after three sessions of OnabotA than patients with CM but without daily headache. In addition, we analysed if the number of headache days per month at baseline was related to the probability and degree of response to OnabotA.

## 2. Results

### 2.1. General Characteristics

During the study period, we included 273 patients. Mean age was 43.9 (SD: 26.7) years and 235 (86.1%) were women. The mean age of migraine onset was 19.0 (SD: 9.3) years and the patients had a mean of 23.5 (SD: 11.8) years with migraine before OnabotA treatment and 34.8 (SD: 45.2) months with CM before OnabotA treatment.

The median number of prior preventive treatments was 3.0 (IQR: 2–4); beta-blockers in 196 (71.8%), neuromodulators in 261 (95.6%), calcium channel blockers in 116 (42.5%), antidepressants in 141 (51.7%), and other treatments in 83 (30.4%) patients.

At baseline, patients had 23.5 (SD: 5.7) headache days per month as the mean. Among them, 120 (44.0%) had 15–21 pain days per month, 64 (23.4%) had 22–29 pain days per month, and 89 (32.6%) had daily headache. Patients had 13.0 (SD: 6.7) severe headache days per month as the mean. Patients attended the emergency department 0.2 (SD: 0.7) days per month, used acute medication 17.8 (SD: 7.9) days, and used triptans 7.0 (SD: 7.2) days per month. Medication overuse criteria was fulfilled by 211 (77.3%) patients.

At first, treatment with OnabotA was started in 305 patients with CM, and 32 (10.5%) patients did not complete three treatment cycles. Between them, nine (28.1%) patients had 15–21 pain days per month, 17 (53.1%) patients had 22–29 pain days per month, and six (18.8%) patients had daily headache. The reasons for discontinuation were follow-up interruption in 13 (40.6%) patients, personal decision not related to lack of improvement in nine (28.1%), lack of improvement in five (15.6%), and poor tolerability in five (15.6%).

All patients of our sample were treated with at least 155 U of OnabotA, 155 U in 110 (40.3%) patients, and >155 U in 163 (59.7%) patients. All patients with >155 U received 195 U.

Among the 273 patients treated with OnabotA, 32 (11.7%) had adverse advents. Between them, 27 (84.4%) patients had injection-site pain, two (6.3%) patients had palpebral ptosis, one (3.1%) patient had syncope, one (3.1%) patient had facial asymmetry, and one (3.1%) patient had dysphagia.

### 2.2. Treatment Response

#### 2.2.1. Total Sample

Between weeks 32–36, the mean number of headache days per month was 12.3 (SD: 9.8) and the mean number of severe headache days per month was 6.4 (SD: 6.4). The mean number of emergency department visits was 0.1 (SD: 0.7) days per month. Patients used acute medication 9.6 (SD: 8.5) days per month as the mean and triptans 4.9 (SD: 6.2) days per month as the mean.

After OnabotA treatment, 166 (60.8%) had a 50% response rate and 81 (29.7%) had a 75% response rate. Considering the number of days, we observed after three sessions of OnabotA treatment a reduction of 11.2 (SD: 9.7) headache days per month, 6.6 (SD: 8.2) severe headache days per month, 8.1 (SD: 10.2) days of acute medication use, and 2.1 (SD: 7.2) days of triptans use per month.

After OnabotA treatment, the 50% response rate and the reduction of headache days per month and severe headache days per month were different within groups (Table 1).

#### 2.2.2. Primary Endpoint

Patients with daily headache had a mean reduction of 14.9 (SD: 9.7) headache days per month, higher than patients with 15–21 and 22–29 headache days per month (*p* < 0.001) (Figure 1).

Regarding response rate, 53 (59.6%) patients with daily headache had a 50% response rate, higher than patients without daily headache (*p* = 0.030). Moreover, 28 (31.5%) patients with daily headache had a 75% response rate. The response rate in each group of patients is described in Figure 2.

### 2.3. Secondary Endpoint

In the univariate regression analysis, per every additional headache day at baseline that patients had, there was a reduction of 0.474 headache days after OnabotA treatment (Figure 3).

In the multivariate analysis, after adjusting by age of migraine onset, years with migraine before OnabotA treatment, months with CM before OnabotA treatment, medication overuse, and number of previous preventive treatments, a higher number of headache days per month remained statistically significant (OR: 0.540, 95% CI: 0.333–0.746, *p* < 0.001). Table 2 shows the results of the univariate and multivariate regression analysis.

## 3. Discussion

In this study, we analysed whether patients with CM and daily headache had a different response who had completed three sessions of OnabotA, than patients without daily headache. Thus, we conducted a prospective cohort study and we evaluated whether the response to OnabotA in patients with CM differed depending on the number of headache days per month at baseline.

The main results of our study were that daily headache was not associated with a worse response to OnabotA and that a higher frequency of headache expressed in number of headache days per month at baseline was associated with a higher response to OnabotA evaluated after three sessions of treatment.

Chronic headaches are grouped together under the descriptive term chronic daily headache (CDH). Despite its name, the term CDH is used to refer to headaches that occur 15 or more days per month for more than 3 months, either by chronic tension-type headache, new daily persistent headache, hemicrania continua, or CM [11].

Patients with 15 or more headache days per month for 3 or more months, of which 8 or more days meet criteria for migraine, will be diagnosed with CM [11]. Between them, up to 33% of patients could suffer from daily headache [10,18]. This would represent the 3.3% of the patients evaluated in the neurology department and 7.7% in headache units [11]. However, the characteristics and response to treatment of this group of patients have hardly been studied [10,11], and the current ICHD does not differentiate between patients with and without daily headache [19].

OnabotA is used in Spain as a preventive treatment for CM since its approval in 2012 [8]. OnabotA is secreted from the *Clostridium botulinum* bacteria [20], and the irreversible blockade of the presynaptic release of acetylcholine at the neuromuscular junction is its best-known effect [21].

Migraine headache trigger sites underlying the mechanism behind the success of extracranial therapies have been widely studied [22]. The main sensory input to the face and head comes from the trigeminal nerve. In CM, the release of inflammatory neuropeptides from stimulated trigeminal sensory neurons is blocked by pericranial injections of OnabotA [21,22]. Thus, the central neurons involved in the perpetuation of pain in CM could be attenuated antidromically [21,23].

OnabotA is an effective, safe, and well-tolerated treatment that reduces the frequency of migraine attacks [24,25], the frequency of moderate-severe pain days, and the hours of headache [5,12,13,26,27]. The PREEMPT trials demonstrated the efficacy and safety of OnabotA as a preventive for 56 weeks treatment in CM [5,12,13]. The COMPEL study confirmed these results and provided data up to 108 weeks [28] and further real-life studies corroborated these previous conclusions [26,29].

However, there is little information about the use of OnabotA in patients with CM and daily headache and this group of individuals has been almost systematically excluded from clinical trials [5,12,13]. In 2019, a retrospective study including patients with CM and MOH concluded that OnabotA was also effective in patients with daily headache. However, the study did not compare the response to OnabotA in patients with daily headache versus those without daily headache [30].

Another prospective observational study, also published in 2019, found that patients with CM and daily headache had a higher reduction in headache frequency than patients without daily headache after OnabotA treatment. Patients with daily headache obtained a mean reduction of 5.7 headache days per month at 24 weeks of treatment and patients that completed 60 weeks of treatment obtained a mean reduction of 8.3 headache days per month [31]. In our study, we decided to analyse results after three procedures, since that is the minimum duration of the treatment, according to the local standard of care [8].

There are discrepancies between authors when describing the headache frequency as a prognostic factor of response to preventive treatment in CM. Some authors have suggested that there could be a better response in patients with a higher number of headache days per month [32], while others hypothesized that daily headache is not associated with the probability of response [33]. The studies that specifically evaluated daily headache, hypothesized at first that its presence could be a predictor of a worse response to preventive treatment in CM [10,18]. However, in recent years, new studies have been published analysing the impact of daily headache in preventive treatment with OnabotA with discordant results. In 2020, a prospective cohort study concluded, after analysing 221 patients, that the patients with daily headache had 3.2 (95% CI: 1.33–7.68, *p* = 0.009) higher odds of 75% response rate to Onabot A at 6 months of treatment, which was not maintained at 12 months [34]. In our study, the presence of daily headache was also a predictive factor of a higher response to OnabotA treatment.

There are several limitations in our study. This is an observational single-center study, without a control group, in which the sample size was not calculated. There could be a selection bias because only patients from a tertiary hospital were included. Considering that all patients should complete at least three sessions with OnabotA due to its cumulative effect, the treatment response was assessed at nine months instead of month-to-month or quarterly. Intention-to-treat analysis was not performed. More research on this subject is needed to confirm the results of our sample.

## 4. Conclusions

Patients with CM and daily headache did not respond worse to OnabotA after three sessions, compared with the rest of patients with CM. In our study, a higher frequency of headache days per month at baseline was associated with a better response to OnabotA.

## 5. Materials and Methods

### 5.1. Study Design and Data Sources

We conducted an observational analytic study with prospective cohort design, following the STrengthening the Reporting of OBservational Studies in Epidemiology (STROBE) statement [35]. The study took place in the headache unit of Valladolid University Hospital, a third-level hospital with a reference population of 280,000 people. We recruited patients between May 2012 (date of OnabotA approval in Spain) and November 2020, using a non-probabilistic sampling method, screening all consecutive patients.

### 5.2. Eligibility Criteria

The inclusion criteria were: (1) age over 18 years old; (2) CM diagnosis according to the International Classification of Headache Disorders (ICHD) operating at the time, either 2nd version [36], 3rd beta version (ICHD-3b) [37], or 3rd version (ICHD-3) [19]; (3) inadequate response or tolerance to at least two oral preventive drugs [8,38]. According to the Spanish Guidelines and the local Standard of Care, all patients received at least three sessions of OnabotA, with at least one session with 195 Units, before considering response as inadequate.

Patients were excluded if they (1) had a serious systemic disease with higher impact than migraine that could be life-threatening; (2) were pregnant or in a breastfeeding status; (3) had hypersensitivity to any of the components of OnabotA; (4) had prior history of severe psychiatric disorders (bipolar and related disorders, dissociative disorders, schizophrenia spectrum disorders, or other psychotic disorders); (5) had prior history of drug abuse or intellectual disability; (6) patients that did not complete three sessions of OnabotA; (7) had a diagnosis of other CDH; chronic tension-type headache, new daily persistent headache, or hemicrania continua.

### 5.3. Procedure

All patients received 31 OnabotA subcutaneous injections of 5 U across seven head and neck muscles (frontalis, corrugator, procerus, occipitalis, temporalis, cervical paraspinal, and trapezius) for a total of 155 U, as indicated by PREEMPT protocol. Depending on the patient’s profile, additional 40 U of OnabotA across three specific muscle groups (occipitalis, temporalis, and trapezius) were permitted, using the “follow-the-pain” approach and reaching the total maximum dose of 195 U in 39 locations [12,13]. The interval between the administration of each treatment session was 12 weeks.

### 5.4. Variables

We analysed demographic variables (sex, age). We also evaluated the age of migraine onset, the number of years with migraine before OnabotA treatment, and the number of months since the onset of CM before OnabotA treatment.

Patients completed a headache diary both before and after starting OnabotA treatment. The baseline number of headache days per month was assessed in the month prior to the start of treatment. We subdivided patients into three groups based on the frequency of headache days per month during baseline period. The first group include patients with 15–21 headache days per month, the second were patients with 22–29 headache days per month, and the third, patients with daily headache (headache 30 or 31 days per month or every day during the last three months). Daily headache was defined as the presence of headache 30 or 31 days per month or every day of February during the last three months. We also recorded the number of severe headache days, defined as days with headache of intensity equal or higher than 7/10 in numeric rating scale, the number of visits to the emergency department, and the number of days per month using symptomatic treatments including triptans.

We defined the presence of medication overuse according to ICHD-3 criteria [19] and the number and type of prior preventive treatments of each patient, according to the national guidelines. We included both treatments with insufficient clinical response and inadequate tolerability. No medication overuse detoxification protocol was used prior to OnabotA administration [39].

We analysed the number of patients who had received at least one session of OnabotA, and we described the reasons for not completing three treatment cycles. All patients of our sample were treated with at least 155 U of OnabotA. We recorded the number of patients treated with 155 U or > 155 U of OnabotA.

We also analysed the frequency and types of adverse events of OnabotA treatment.

### 5.5. Endpoints

#### 5.5.1. Primary Endpoint

The primary endpoint of the study was to assess whether patients with CM and daily headache had a different response to OnabotA treatment than CM patients without daily headache between weeks 32–36, compared with the baseline.

For this purpose, we compared the headache frequency of the patients at baseline with the headache frequency after treatment with OnabotA. Given the cumulative effect of OnabotA over at least three treatment cycles [5,8,12,13], response was assessed between weeks 32–36.

To compare the treatment response, we used two methods, both of them based on the Guidelines of the International Headache Society for controlled trials of preventive treatment of CM in adults [40]. We used 50% response rate and 75% response rate, defined as at least 50% or 75% reduction in the number of headache days from baseline [40]. We calculated the proportion of patients with a 50% and 75% response rate between weeks 32–36, compared with the baseline.

#### 5.5.2. Secondary Endpoints

The secondary endpoint was to analyse the presence of response predictors that could be associated with a higher response to OnabotA treatment. We conducted a linear regression analysis, and we included in the multivariate analysis the variables that had been described to be related with the probability of response to OnabotA: duration of migraine [41,42] and medication overuse [43,44]. We also included in the analysis the number of previous preventive treatments, as we considered it a clinically relevant variable.

Finally, we compared the reduction in the use of symptomatic treatments, triptans, and the reduction in number of emergency department visits per month after three sessions of OnabotA in patients with CM and daily headache and patients without daily headache.

### 5.6. Ethical Aspects

The local research ethics committee (PI: 17-832 and PI: 19-1460) approved this study on 26 October 2017. Written informed consent for participation was obtained from all patients. This study was conducted in accordance with the principles of the Declaration of Helsinki.

### 5.7. Statistical Analysis

Qualitative and ordinal variables were expressed as frequency and percentage. Quantitative continuous variables were presented as mean and standard deviation (SD) if the distribution was normal and as median and interquartile range (IQR) otherwise. We evaluated if the distribution was normal by using Kolmogorov–Smirnov test. We employed Chi2 test or Fisher’s exact test to compare qualitative variables. For hypothesis testing, we used Student *t*-test or analysis of variance when the distribution was normal, and Mann–Whitney U test or Kruskal–Wallis otherwise.

For the study endpoints, we conducted a univariate linear regression analysis of all baseline variables, and all the variables that showed statistical association with *p* < 0.2 were included in a multivariate linear regression analysis. We present the odds ratio (OR) and the 95% confidence interval (CI). The level of significance threshold was set in 0.05 and we used Bonferroni method for multiple comparisons correction. We did not calculate sample size in advance. Statistical analysis was performed with SPSS v.26 (IBM Corp. Armonk, NY, USA).

## Figures and Tables

**Figure 1 toxins-13-00432-f001:**
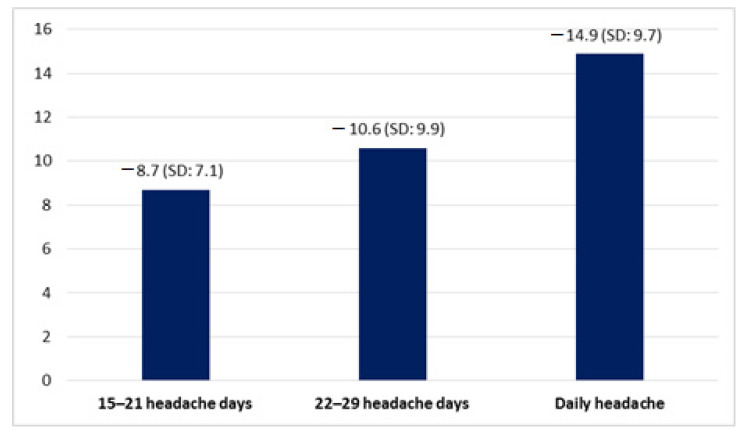
Reduction in number of headache days per month.

**Figure 2 toxins-13-00432-f002:**
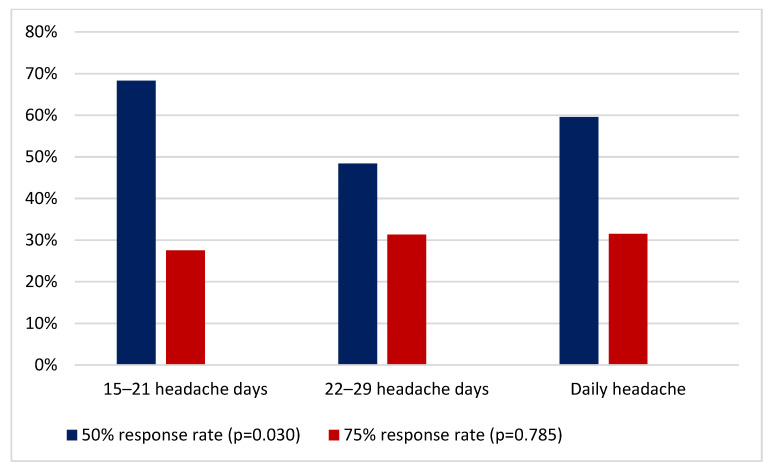
Response rate in each group of patients.

**Figure 3 toxins-13-00432-f003:**
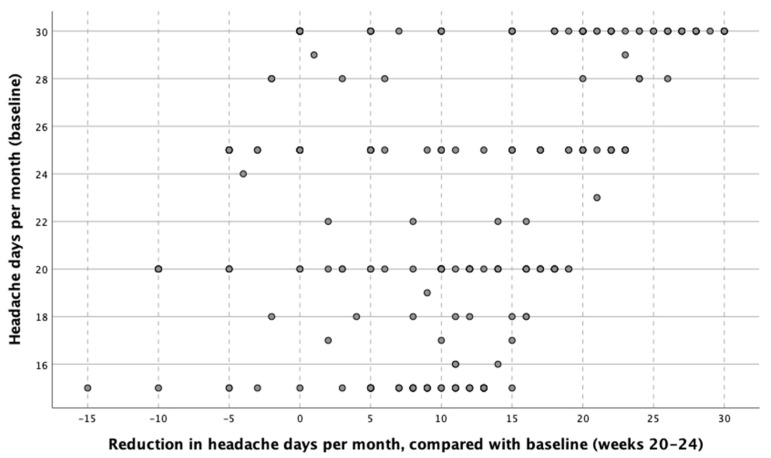
Reduction of headache days per month between weeks 32–36 after three sessions of OnabotA, based on the number of headache days per month at baseline.

**Table 1 toxins-13-00432-t001:** Response to treatment with OnabotA in CM patients with and without daily headache.

Variable	Total Sample (*n* = 273)	Patients with 15–21 Headache Days/Month (*n* = 120)	Patients with 22–29 Headache Days/Month (*n* = 64)	Patients with Daily Headache (*n* = 89)	*p*-Value
50% response rate (N)	166 (60.8%)	82 (68.3%)	31 (48.4%)	53 (59.6%)	(*p* = 0.030)
75% response rate N (%)	81 (29.7%)	33 (27.5%)	20 (31.3%)	28 (31.5%)	(*p* = 0.785)
Reduction of headache days per month (days)	11.2 (SD 9.7)	8.7 (SD: 7.1)	10.6 (SD: 9.9)	14.9 (SD: 9.7)	(*p* < 0.001)
Reduction of severe headache days per month (days)	6.6 (SD: 8.2)	4.9 (SD: 6.5)	7.3 (SD: 7.6)	8.4 (SD: 10.2)	(*p* = 0.008)
Reduction of days of acute medication days use per month (days)	8.1 (SD: 10.2)	6.0 (SD: 7.9)	8.8 (SD: 11.2)	10.4 (SD: 11.6)	(*p* = 0.006)
Reduction of days of triptan use per month (days)	2.1 (SD: 7.2)	2.2 (SD: 7.1)	3.8 (SD: 7.3)	0.8 (SD: 7.3)	(*p* = 0.040)

**Table 2 toxins-13-00432-t002:** Predictors of response to treatment with OnabotA. Univariate and multivariate regression analysis.

Variable	Analysis	OR	IC 95%	*p*-Value
Age at the time of the first OnabotA session	Univariate Multivariate	−0.017	(−0.060)–(0.026)	0.436
Age of migraine onset	Univariate Multivariate	0.095 0.080	(−0.028)–(0.218) (−0.064)–(0.224)	0.130 0.275
Years with migraine before OnabotA treatment	Univariate Multivariate	−0.025 −0.007	(−0.123)–(0.073) (−0.122)–(0.108)	0.615 0.902
Months with CM before OnabotA treatment.	Univariate Multivariate	−0.013 −0.033	(−0.039)–(0.012) (−0.059)–(0.007)	0.312 0.013
Sex	Univariate Multivariate	−1.040	(−4.374)–(2.294)	0.540
Headache days per month	Univariate Multivariate	0.474 0.540	0.278–0.670 0.333–0.746	*p* < 0.001 *p* < 0.001
Medication overuse	Univariate Multivariate	0.454 −0.509	(−2.302)–(3.210) (−3.225)–(2.206)	0.746 0.712
Number of previous preventive treatments	Univariate Multivariate	−0.047 −0.012	(−0.766)–(0.673) (−0.703)–(0.678)	0.898 0.972

## Data Availability

The datasets used and/or analysed during the current study are available from the corresponding author on reasonable request.

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
