# Peer review of "Daily Headache in Chronic Migraine Is a Predictive Factor of Response in Patients Who Had Completed Three Sessions of OnabotulinumtoxinA"

_toxins, 2021, doi:10.3390/toxins13060432_

Round 1

Reviewer 1 Report

The results are new, the topic interesting to a global audience. The following reference is missing: 

Gfrerer L, Raposio E, Ortiz R, Austen WG. Surgical treatment of migraine headache: Back to the future. Plast Reconstr Surg. 142(4): 1036-45, 2018.

Reviewer 2 Report

This is a nicely written paper on onabotulinumtoxin A in patients with chronic migraine including subjects with daily headache.

The major shortcoming is the exclusion of patients who did not complete three treatment cycles. The title of the paper is not supported by the study, instead the authors should clearly state that the response to treatment with onabotulinumtoxin A in chronic migraine is restricted to patients who had completed three treatment cycles.

In addition, the authors should add information about all patients who had received at least one treatment with onabotulinumtoxin A and they should give the number of patients who did not complete three treatment cycles separately for the three study groups. They should also give the reasons for non-completing three treatment cycles.

Moreover, details of treatment with onabotulinumtoxin A should be added, i.e. the interval from first to second and from second to third treatment and the number of subjects treated with 155, <155 and >155 units in each of the three treatment cycles, separately for the three study groups.

Finally, information about adverse events should be added.

Minor points:

Line 28: Ref. 1 does not refer to chronic migraine. The statement requires revision.

Lines 29-30. The phrase “In contrast” is out of place in the sentence: “In contrast to other disabling neurological diseases, migraine treatment can improve patient´s quality of live,…” (correct: life)
